# Molecular Dynamics Simulation of Cumulative Microscopic Damage in a Thermosetting Polymer under Cyclic Loading

**DOI:** 10.3390/polym16131813

**Published:** 2024-06-26

**Authors:** Naoki Yamada, Mayu Morita, Maruri Takamura, Takahiro Murashima, Yutaka Oya, Jun Koyanagi

**Affiliations:** 1Department of Materials Science and Technology, Graduate School of Advanced Engineering, Tokyo University of Science, Tokyo 125-8585, Japan8217056@alumni.tus.ac.jp (M.T.); koyanagi@rs.tus.ac.jp (J.K.); 2Department of Physics, Graduate School of Science, Tohoku University, Sendai 980-8578, Japan; murashima@tohoku.ac.jp

**Keywords:** molecular dynamics, thermosetting resin, mechanical properties, cyclic loading

## Abstract

To develop durable composite materials, it is crucial to elucidate the correlation between nanoscale damage in thermosetting resins and the degradation of their mechanical properties. This study aims to investigate this correlation by performing cyclic loading tests on the cross-linked structure of diglycidyl ether bisphenol A (DGEBA) and 4,4′-diaminodiphenyl sulfone (44-DDS) using all-atom molecular dynamics (MD) simulations. To accurately represent the nanoscale damage in MD simulations, a bond dissociation algorithm based on interatomic distance criteria is applied, and three characteristics are used to quantify the microscopic damage: stress–strain curves, entropy generation, and the formation of voids. As a result, the number of covalent bond dissociations increases with both the cyclic loading and its amplitude, resulting in higher entropy generation and void formation, causing the material to exhibit inelastic behavior. Furthermore, our findings indicate the occurrence of a microscopic degradation process in the cross-linked polymer: Initially, covalent bonds align with the direction of the applied load. Subsequently, tensioned covalent bonds sequentially break, resulting in significant void formation. Consequently, the stress–strain curves exhibit nonlinear and inelastic behavior. Although our MD simulations employ straightforward criteria for covalent bond dissociation, they unveil a distinct correlation between the number of bond dissociations and microscale damage. Enhancing the algorithm holds promise for yielding more precise predictions of material degradation processes.

## 1. Introduction

Carbon fiber–reinforced plastic (CFRP) possesses excellent specific strength and specific stiffness and has been widely applied in the aerospace field, especially in large transport aircraft [1,2]. CFRP comprises a complex structure consisting of carbon fibers and thermosetting resin and exhibits various forms of damage and failure under mechanical loading (such as fiber breakage, transverse cracks, matrix cracks, etc.). To predict the degradation of mechanical properties caused by these types of damages, various computer simulations have been applied to composite materials.

Among the various computational approaches, the finite element method (FEM) and molecular dynamics (MD) simulations are prominent for addressing the damage mechanisms of polymers and polymer composites. While FEM has successfully reproduced failure and mechanical properties quantitatively [3,4,5], the MD simulations used in this study offer certain advantages. MD simulations calculate thermodynamic properties based on atomic motion, whereas the FEM relies on the constitutive law of a material expressed as a continuum body. Consequently, MD simulations allow us to examine the dependence of microscopic damage on material degradation at the atomic scale.

In this study, MD simulations were performed to detect microscopic damage to the thermosetting resin that constitutes the matrix portion of CFRP. The thermosetting resin forms a three-dimensional cross-linked structure with covalent bonds, ensuring a lighter weight, superior mechanical properties, and higher thermal and chemical resistance [6]. Therefore, thermosetting resins are widely used not only in large structural components but also in automotive parts, industrial product components, and electrical and electronic products [7,8]. However, cross-linked structures are susceptible to damage under mechanical loading conditions, resulting in brittle fractures at lower strain levels compared to thermoplastic polymers [9,10]. Improving the durability of CFRP fundamentally relies on identifying nanoscale damage modes under cyclic loading conditions that mimic their usage as structural materials [11]. MD simulation is thought to be an appropriate approach to address this issue.

Although MD simulation has been applied for polymers [12,13,14,15], reinforcements [16,17], and their composites [18,19,20,21,22,23,24,25,26,27,28,29,30], many studies have been limited to analyzing mechanical properties near equilibrium states, such as Young’s modulus, density, and glass transition temperature. This is because MD simulations traditionally assume that the topology of the individual molecules does not change over time (such an MD method is hereafter referred to as classical MD), leading to difficulty in reproducing the damage associated with covalent bond dissociation.

Recently, MD simulations using the reactive force field (Reax-FF) have drawn attention due to their ability to smoothly represent the formation and dissociation of covalent bonds by approximating the bond order as a continuous function of interatomic distance. Several studies using Reax-FF have explored mechanical properties over a wide range of strains that are difficult to achieve with classical MD methods [31,32,33]. However, Reax-FF is computationally demanding and often less quantitative in its mechanical properties than classical MD. Therefore, in this study, we introduced a distance-based bond-breaking criterion into classical MD simulations. This criterion is the simplest approach when considering bond breaking. However, it has been widely used in many MD simulations involving topology changes such as chemical reaction calculations [34,35,36,37,38,39,40,41,42,43,44,45,46,47], making it highly relevant for adoption.

Previous studies have also explored the durability testing of thermosetting polymers. For example, Park et al. and Li et al. investigated the correlation between changes in the internal structure of thermosetting resins and their mechanical properties using classical MD calculations that did not involve covalent bond dissociation [48,49]. Similarly, Schichtel et al. conducted cyclic loading tests considering both bond formation and dissociation using Reax-FF [50]. They failed to accurately reproduce the fracture strain and fracture stress of real thermosetting resins, although they obtained valuable insights into the changes in the internal structure of thermosetting polymers. Our methodology, integrating cyclic loading calculations and covalent bond dissociation within classical MD simulations, represents an important step forward in quantitatively assessing microscale damage, introducing a unique novelty absent in prior studies.

At the end of the Section 1, the quantification of microscale damage is described. We focus on the degradation of mechanical properties, entropy generation, and void content inside the system. Entropy is expected to serve as a physical quantity for quantitatively measuring the degradation of materials, such as the difficult-to-measure bond dissociation and void content in experiments [51,52,53,54]. Therefore, verifying the correlation between entropy and bond dissociation or void content is highly significant for assessing the remaining life of CFRP.

In response to the above background, the purpose of this study is to reproduce the cyclic loading test for thermosetting polymers and investigate microscopic damage using entropy and voids in classical MD simulations while considering covalent bond dissociation. The remainder of this paper is organized as follows: The Section 2 describes the simulation method and simulation system, including the calculation conditions of cyclic loadings and the algorithm for covalent bond dissociation. The Section 3 presents the simulation results and discussions. Finally, the results are provided in the Section 4.

## 2. Simulation Method

In this study, we reproduced the chemical reactions of thermosetting resin to create a cross-linked structure, followed by conducting tensile and cyclic loading calculations on that structure to quantitatively evaluate material damage. In this section, we present the chemical structure of the resin, outline the calculation algorithm, and specify the computational conditions. All MD calculations in this study were performed using a Large-scale Atomic/Molecular Massively Parallel Simulator (LAMMPS) [55], with the aid of a supercomputer at the Institute for Solid State Physics, University of Tokyo.

### 2.1. Molecular Structure Construction

In this study, we utilized diglycidyl ether bisphenol A (DGEBA) as the base resin and 4,4′-diaminodiphenylsulphone (44-DDS) as the curing agent, typically used in aerospace-grade thermosetting resins. Their chemical structures are illustrated in Figure 1. DGEBA features an epoxy group at each terminus of its molecular structure, while the curing agent presents primary amine groups at both ends. These components react to form a cross-linked structure in the system. To construct the three-dimensional topology data of these molecules, we employed Marvin Sketch and Open Babel. Subsequently, force field parameters and charges were assigned using Polypergen [56]. We utilized the Optimized Potentials for Liquid Simulations (OPLS-AA) force field for molecular structure and assigned ESP charges to each atom through density functional theory calculations based on the B3LYP/6-31G (Hamiltonian/basis set) [57,58,59].

### 2.2. Curing Simulation

We created equilibrium structures of the two-component multimolecular systems consisting of DGEBA and 44-DDS, followed by chemical reaction calculations to generate their cross-linked structures. DGEBA and 44-DDS were introduced into the system so that the total atomic number reaches approximately 27,000. The number ratio of base resin to curing agent was set to the stoichiometric ratio of functional groups, ensuring all functional groups reacted without excess or deficiency. A stepwise relaxation was then performed to create an equilibrium structure at a constant temperature of T = 300 K and atmospheric pressure (P = 1 atm) before the cross-linking reaction.

To ensure a uniform cross-linked structure, it is essential that the molecules are uniformly dispersed prior to the cross-linking reaction. Since DGEBA and 44-DDS are in liquid and solid states, respectively, at room temperature, the molecules were first randomized at a high temperature. Thus, relaxation calculation was performed in the NVT ensemble at a constant temperature of T = 600 K to randomize the molecular arrangement inside the system. Subsequently, relaxation calculations in the NVT ensemble and contraction of the system box were repeatedly conducted to increase the density to around 0.8 g/cc. Finally, relaxation calculation for 5 ns was performed in the NPT ensemble (T = 300 K, P = 1 atm) to obtain the equilibrium structure. This equilibrium structure was used as the initial state for chemical reaction calculations.

As shown in Figure 2a, a primary amine reacts with an epoxy group to form a secondary amine, and a secondary amine reacts with an epoxy group to form a tertiary amine. The tertiary amine does not undergo further reactions with epoxy groups. Through such sequential reactions, branching structures are formed via covalent bonds within the system, resulting in a dense cross-linked structure. In this study, such chemical reaction processes were reproduced with a reaction algorithm based on functional group distance criteria, as described below.

During the NPT relaxation calculation, all pairs of an oxygen atom at epoxy groups and a nitrogen atom at primary or secondary amines within a 5 Å distance were listed. The topology of these molecules was then changed to match the topology of the reaction process depicted in Figure 2a. The molecules after the reaction underwent a brief NVE relaxation calculation to stabilize the intramolecular potential energy. The reaction calculation was terminated if no chemical reactions occurred within 1 ns. Subsequently, a 5 ns NPT relaxation calculation was performed to equilibrate the cross-linked structure.

The final cross-linked structure, as obtained from the chemical reaction calculations, is depicted in Figure 2b. The degree of curing conversion, i.e., the number of reacted epoxy groups divided by the total number of epoxy groups before the reaction, finally reached 0.83. It is known that when the degree of cure exceeds approximately 0.6, all monomers of the base resin and curing agent have reacted to form a single molecule, that is, the dense cross-linked structure. Previous studies have also indicated a degree of curing conversion of around 0.8 [30,31,32], suggesting that the obtained structure is reasonable.

### 2.3. Cyclic Loading Simulation

Cyclic loading was conducted on the cross-linked structure obtained in the previous section. The system underwent a repeated uniaxial extension and contraction at a constant strain rate of 4 × 10^9^/s and temperature of T = 300 K (room temperature). In general, the strain rates used in MD simulations are much higher than those in experiments. The strain rate used in this study was determined based on the previous research by Park et al., where two different strain rates were employed for cyclic loading simulations [49]. We selected a strain rate slightly lower than the lower rate used by Park et al. Three different strain amplitudes were applied: 0–0.02, 0–0.03, and 0–0.04, all of which are below the fracture strain range of actual thermosetting resins under uniaxial tensile tests. A constant pressure of 1 atm was applied perpendicular to the tensile direction to mimic Poisson contraction.

During the cyclic loading tests, we introduced covalent bond dissociation based on distance criteria. Figure 3 illustrates the potential covalent bonds that may break in the molecular structure. Single bonds, excluding benzene rings, epoxy groups, and primary amines, are considered candidates for bond breaking. Additionally, a covalent bond connecting the nitrogen atom of secondary amines to a carbon atom is also considered a potential bond-breaking candidate. In this study, each covalent bond was set to break when its length exceeded 1.1 times the equilibrium bond length ($\varepsilon_{0}$) for the OPLS-AA force field. Preliminary validation using the uniaxial tensile tests was conducted to investigate the relationship between the criteria length and the strain at which covalent bond dissociation begins. The results showed that bond dissociation initiated at a strain of approximately 5% when $\varepsilon_{0}$ was used. This value was set as the criteria length, ensuring that no covalent bond dissociation occurred up to the maximum strain amplitude of 4% used in this study. While covalent bond dissociation was unlikely to occur in the uniaxial tensile tests under our parameter settings, it is crucial to acknowledge the potential for dissociations if the cross-linked structure undergoes substantial changes from its equilibrium state due to cyclic loading. After the bond breaking, a brief relaxation calculation (10 fs) was performed under NVT ensemble conditions (T = 300 K) to reduce the forces exerted on the atoms constituting the broken bond. This study notes that a small-time step of 0.01 was used for calculation stability.

In this study, independent cyclic loading calculations were conducted along the x, y, and z directions. The error bars shown in the results for voids, entropy, and the number of covalent bond dissociations represent the standard deviation obtained from these three calculations.

## 3. Results and Discussions

Figure 4 depicts the stress–strain curves for the first and 10th cyclic loadings. In Figure 4a–c, increasing the amplitude of the cyclic strain and the number of cyclic loadings results in the degradation of the mechanical properties; that is, both stiffness and strength decrease, and the cross-linked structure behaves inelastically rather than elastically. Furthermore, comparing Figure 4c with Figure 4d shows that the mechanical properties underwent greater degradation due to covalent bond dissociation.

Figure 5 illustrates the generated entropy used to quantitatively evaluate the degradation of the material. The entropy generated per cyclic load is calculated by dividing the total work by the temperature [51]. Since the temperature is constant at T = 300 K, the entropy is almost proportional to the area surrounded by the hysteresis loop. Note that the vertical axis represents the sum of all entropies generated up to the cycle number on the horizontal axis. This figure shows that the total generated entropy increases as the amplitude of the cyclic strain increases. Covalent bond dissociation also generates more entropy than in the absence of bond dissociation. These entropy differences are more pronounced in the latter phases of the cyclic loading compared to the initial stages. As previous studies have indicated, our results suggest a positive correlation between entropy generation and polymer material degradation [15].

To examine the molecular-scale mechanisms behind this increase in entropy, Figure 6 depicts the correlation between the total number of dissociated covalent bonds and the cyclic loading frequency. This figure shows an increase in the number of covalent bonds dissociated with both the strain amplitude and the number of cyclic loadings, which follows the same trend as the entropy difference between the cases with and without covalent bond dissociation. Therefore, the number of covalent bond dissociations is also positively related to the generated entropy.

Here, we discuss the microscopic mechanisms of material degradation under cyclic loading. In conclusion, it is believed that cross-linked polymers degrade through the following consecutive stages: Initially, during the early stages of cyclic loading, the covalent bonds reorient themselves in the direction of the load. Secondly, these bonds become too unstable to sustain stronger loads and eventually rupture. Thirdly, the load supported by the broken bonds is redistributed among the surrounding covalent bonds. Finally, multiple bonds dissociate, leading to the formation of large voids. In the following, we present results that support this scenario.

Figure 7 shows the difference in the cross-sectional area between the initial system and the system after 10 cyclic loadings. These areas correspond to planes perpendicular to the loading direction, thereby representing the degree of the Poisson effect. In the absence of covalent bond dissociation, the cross-sectional area decreases independent of the strain amplitude, which is consistent with previous MD simulation results [48,49]. The preceding research suggests that the reduction in the area arises from the alignment of covalent bonds parallel to the loading direction. On the other hand, if covalent bond dissociation is present, the area increases with strain amplitude, which clearly correlates with the number of covalent bonds that are broken. These results suggest that the covalent bonds are aligned in the early stages and then broken in the later stages of cycle loading. Our findings indicate that the reduction in the system density correlates with the number of dissociated bonds, signifying an expansion in the system volume pre- and post-cyclic loadings. This phenomenon can be attributed to the following: When covalent bonds break, the atoms within them gain mobility, leading to an increase in the entropy of the center-of-mass freedom. To maximize the entropy, the freedom of movement for polymer end atoms is enhanced, resulting in an expansion of system volume and, consequently, a decrease in density. This increase in volume can be represented as an increase in voids. Therefore, we investigated the relationship between the number of cycles and voids. Here, it is noted that when the strain amplitude is 0.02, as depicted in Figure 7, there is little covalent bond dissociation. As a result, the cross-sectional area decreases, and the system density increases before and after cyclic loading, regardless of the presence or absence of covalent bond dissociation.

The spatial distribution of the voids was determined by estimating the volume of atom-free spaces using the Voronoi tessellation. These calculations were performed using the VORO++ package [60]. Figure 8 shows the size of the maximum voids and total void content with respect to the number of cyclic loadings. When covalent bond dissociation is accounted for, both the maximum void size and the total void volume increase with the number of cycles and strain amplitude. Conversely, when not considered, there is no change in the voids. These trends in void size and void content are essentially the same as the trends in entropy generation (Figure 5), dissociative bond number (Figure 6), and cross-sectional area (Figure 7). Furthermore, the association between the maximum void size and void content suggests that voids are expanding through a mechanism like nucleation growth, which is confirmed by Figure 9 and Figure 10.

Figure 9 illustrates a comparison of void size frequency distributions between the initial system and the system after 10 cyclic loadings. This figure indicates that large voids emerge instead of a decrease in small voids. Figure 10 shows the spatial distribution of voids at each stage of the cyclic loading. In the initial state, small voids are uniformly distributed inside the system. Subsequently, as the number of cycles increases, larger voids are locally formed. These results suggest that covalent bond dissociation propagates locally, causing voids to enlarge by incorporating nearby small voids.

Here, we introduce a previous experimental study on cyclic loading for the same thermosetting polymer as that used in this study. Kudo et al. reported that the density decreases after 100 cyclic loadings [61], which is qualitatively consistent with our results (Figure 7). This decrease in density cannot be reproduced with standard MD simulations without considering bond dissociation [48,49]. Additionally, they evaluated the entropy generation before and after 100 cyclic loadings for an amplitude of 0.01. They estimated a generated entropy of 18.1 kJ/m^3^ through the changes in the heat capacity. A direct comparison between their results and ours is not possible due to the differences in the number of cycles and amplitude. However, for 10 cycles with an amplitude of 0.02, our calculated entropy generation is approximately 7.5 kJ/m^3^. Considering the dependencies of entropy on the amplitude and the number of cycles, we believe there is no significant difference. They also reported an increase in entropy generation per cycle with the number of cyclic loadings, a trend not observed in our study or any previous ones. However, our simulations indicate that considering bond dissociation suppresses the decrease in entropy generation per cycle. These results highlight the importance of bond dissociation in replicating the failure of thermoset polymers under cyclic loading conditions.

Therefore, our computational method, which integrates covalent bond dissociation into classical MD simulations, demonstrates superior density reproducibility compared to conventional classical MD. Additionally, it may offer advantages in computational speed and quantitative accuracy over Reax-FF-based MD simulations. In future perspectives, there is a need to refine the algorithms regarding covalent bond dissociation. While this study employed a uniform threshold of 1.1 times the equilibrium bond length for all types of covalent bonds, ideally, this threshold should vary depending on the bond type. Incorporating the ease of dissociation for each specific bond type requires leveraging quantum chemistry calculations, leading to more quantitative predictions of microscopic damages for various thermosetting polymers and their composite materials. Furthermore, it is important to investigate the long-term durability of thermosetting polymers through tests with a higher number of cyclic loadings. For this purpose, the stability of our MD simulations also needs to be improved.

## 4. Conclusions

The accumulation of microscopic damage induced by cyclic loading was evaluated using all-atom molecular dynamics simulations that incorporated covalent bond dissociation based on interatomic distance criteria. Our results led to the following conclusions:(1)Entropy and void volume increase with both the number of cyclic loadings and their amplitude, resulting in the thermosetting polymer exhibiting inelastic behavior. The accumulated entropy and void volume showed a strong positive correlation with the number of dissociated bonds.(2)The damage at the nanoscale initiates and propagates through the following process: Initially, under cyclic loading, covalent bonds align with the applied load direction. Subsequently, tensioned covalent bonds break, causing the surrounding bonds to fracture sequentially, resulting in the formation of significant voids. Consequently, stress–strain curves display non-linear and inelastic behavior.(3)Covalent bond dissociation reduces the system density and mitigates the decrease in dissipation energy per cycle, qualitatively in line with the previous experimental observations.

In summary, incorporating covalent bond dissociation into molecular dynamics simulations is crucial for reproducing damage in thermosetting resins under cyclic loading. In the future, the development of a multiscale simulation model linking quantum chemistry calculations and FEM through the current MD simulations is expected. This model could accelerate the development of thermosets with long-term durability in polymer composites.

## Figures and Tables

**Figure 1 polymers-16-01813-f001:**
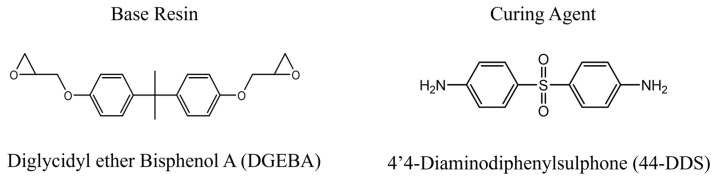
Chemical structures of DGEBA and 44-DDS.

**Figure 2 polymers-16-01813-f002:**
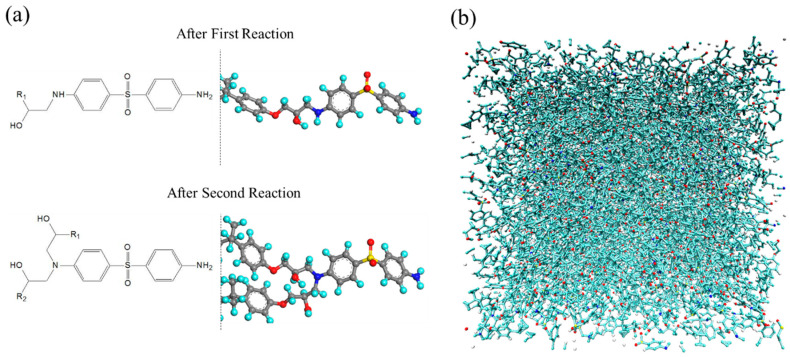
(**a**) Molecular structures after first and second chemical reactions, and (**b**) an equilibrium cross-linked structure obtained from the curing simulation. The red circles denote oxygen atoms, the light blue circles denote hydrogen atoms, the blue circles denote nitrogen atoms, the yellow circles denote sulfur atoms, and the gray circles denote carbon atoms.

**Figure 3 polymers-16-01813-f003:**
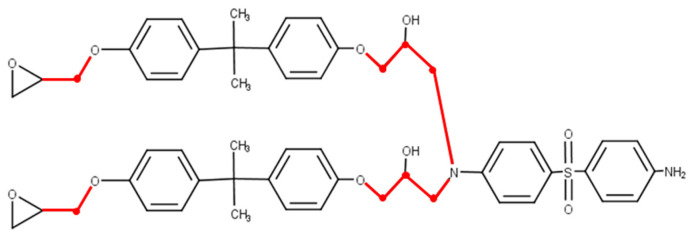
Covalent bonds that potentially break in load simulation are represented by red lines.

**Figure 4 polymers-16-01813-f004:**
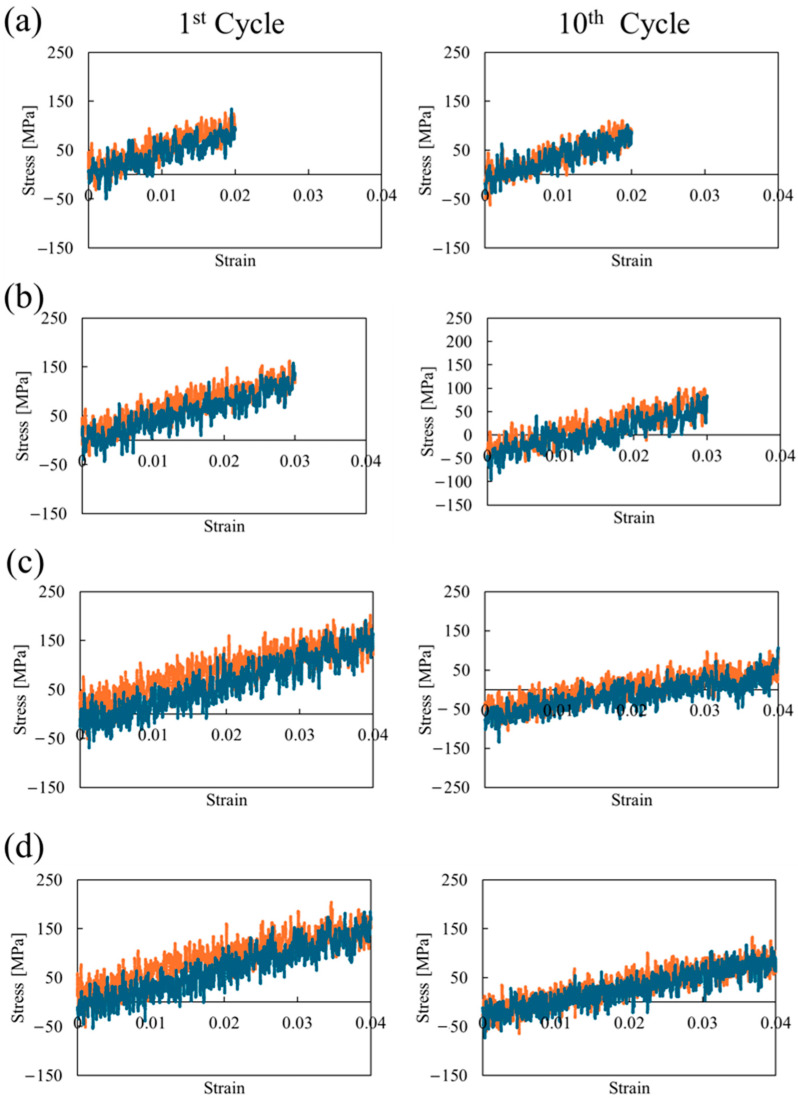
(**a**–**c**) Stress–strain curves for the strain amplitudes of 0.02, 0.03, and 0.04, respectively, considering the dissociation of covalent bonds. (**d**) Shows the case of 0.04 maximum strain without considering covalent bond dissociation. Orange dots represent loading from 0 to 0.04 strain, while blue dots are unloading from 0.04 to 0 strain.

**Figure 5 polymers-16-01813-f005:**
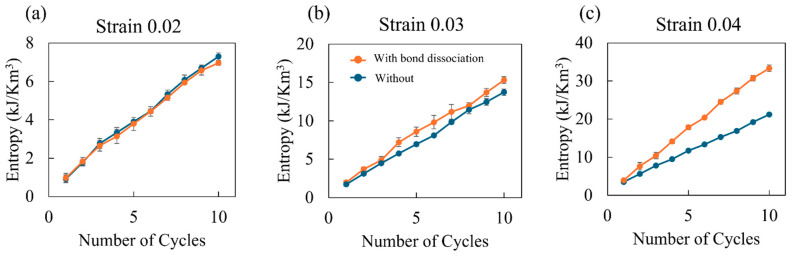
Total entropy generations depending on the number of cycles for strain amplitude of 0.02 (**a**), 0.03 (**b**), and 0.04 (**c**). Orange dots and blue dots are for the cases with and without bond dissociation, respectively.

**Figure 6 polymers-16-01813-f006:**
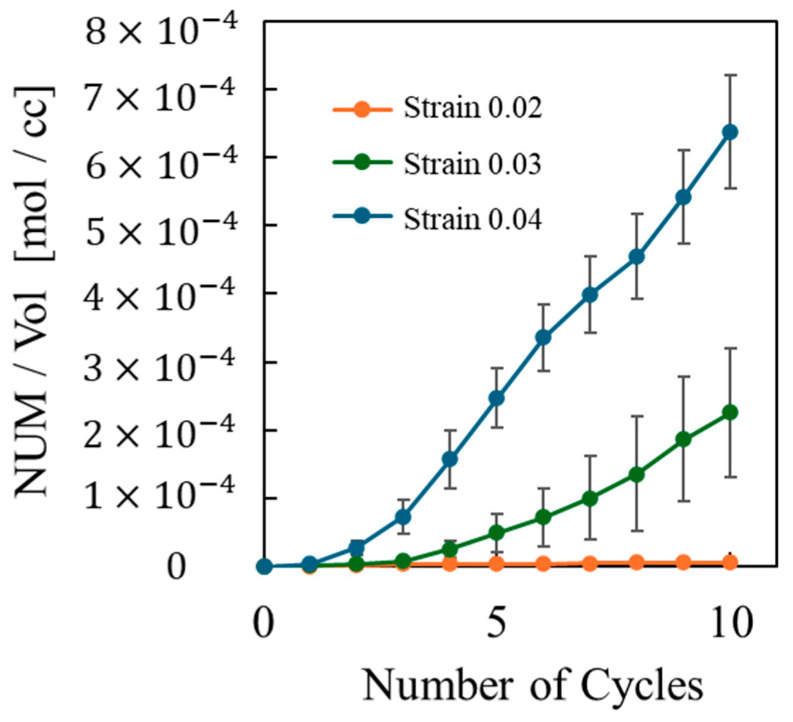
Total number of dissociated bonds depending on the number of cycles for strain amplitude of 0.02 (orange dots), 0.03 (green dots), and 0.04 (blue dots).

**Figure 7 polymers-16-01813-f007:**
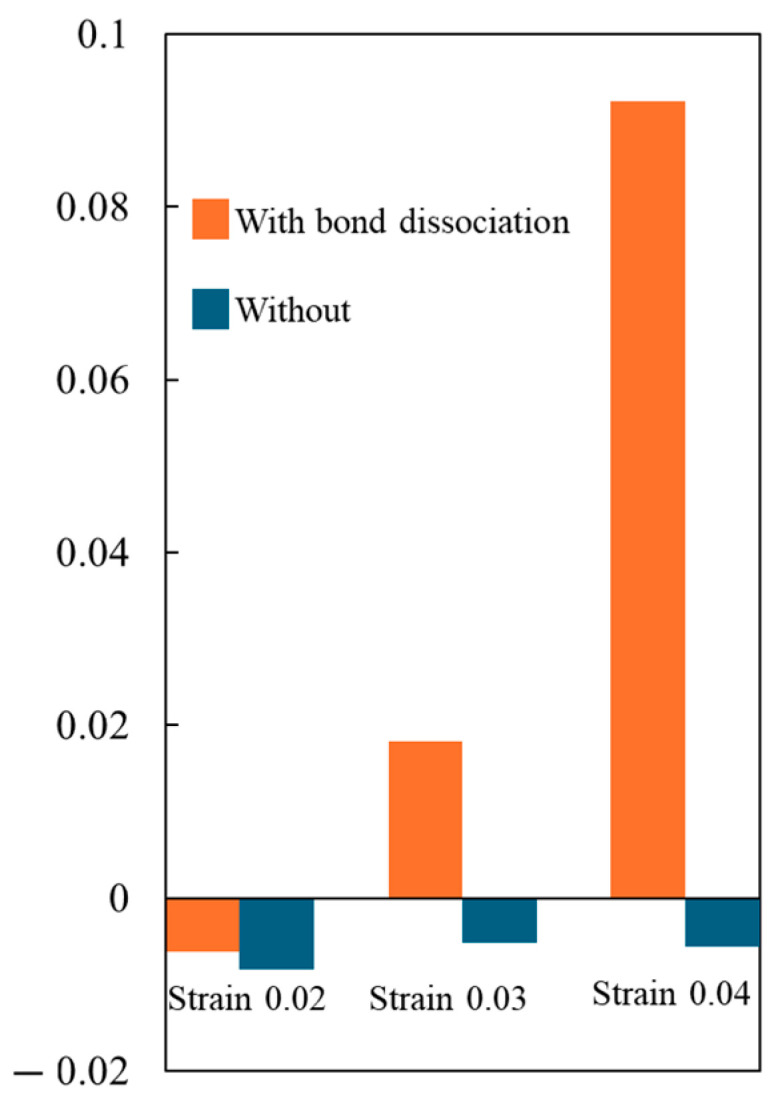
The rate of change in cross-sectional area from the initial state to after 10 cycle loadings.

**Figure 8 polymers-16-01813-f008:**
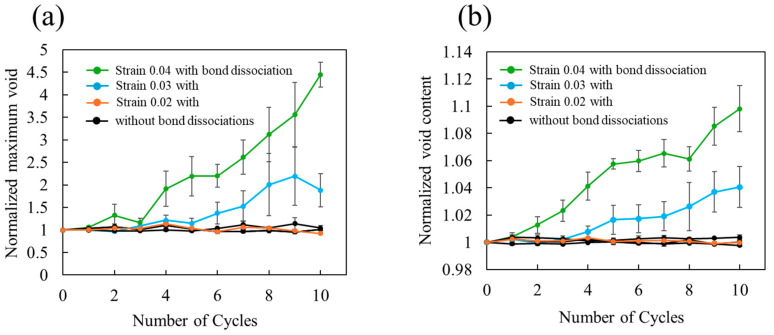
Cycle-number dependence of (**a**) maximum void volume and (**b**) total void content normalized by those of initial state. The green, blue, and orange dots represent strain amplitude of 0.04, 0.03, and 0.02, respectively, accounting for bond dissociation. The three black lines correspond to the same three strain amplitudes for the case of no covalent bond dissociation.

**Figure 9 polymers-16-01813-f009:**
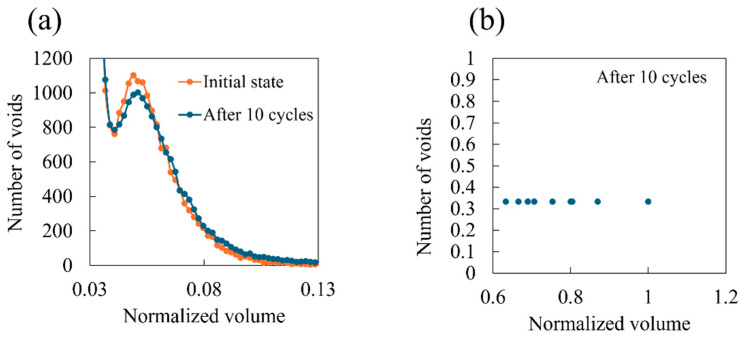
Frequency distributions for void volume normalized by the largest void volume after 10 cycles. (**a**) The small void distribution and (**b**) large void distribution. The orange dots represent the system in its initial state, while the blue dots depict the system after conducting 10 cyclic loadings with a strain amplitude of 0.04, considering the covalent bond dissociation. These are averages for three independent simulation runs.

**Figure 10 polymers-16-01813-f010:**
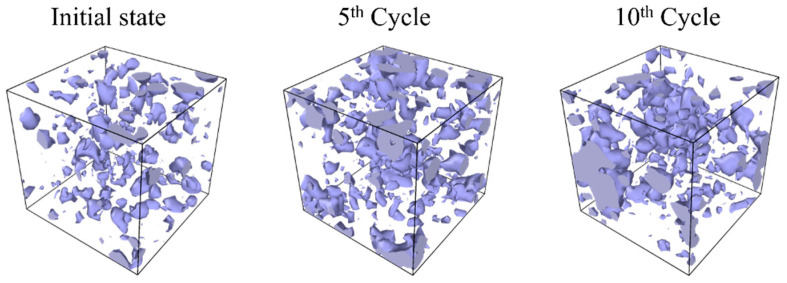
Snapshots of spatial void distributions at each stage of cyclic loadings for strain amplitude of 0.04, considering covalent bond dissociation. Purple areas depict voids.

## Data Availability

Data are contained within the article.

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
