# Peer review of "Molecular Dynamics Simulation of Cumulative Microscopic Damage in a Thermosetting Polymer under Cyclic Loading"

_polymers, 2024, doi:10.3390/polym16131813_

Round 1

Reviewer 1 Report

Comments and Suggestions for Authors

This study has attempted to evaluate the microscopic damage of thermosetting resin under cyclic loading by MD. It seems to be a timely and appropriate study. The paper contains original results and deserves to be published after major revision. In my humble opinion, there is a dire need for providing more data and in-depth discussion.

v Title

It is suggested to include potential application(s) in this section.

v General comment

The authors should use abbreviations after addressing them in the text.

There are several grammatical and typo errors in the text.

The dots at the end of sections’ titles should be removed.   

v Abstract

1.     This section should be re-written. It is highly recommended to follow the standard protocol for writing “abstract”. One sentence to state the importance of the topic, followed by the explicit illustration of the aims (and the variable), employed methods, main achievements, and an outlook in the light of obtained results.

2.     The authors are recommended to add more quantitative data in this section.

3.     The type of “base resin” and “curing agent” should be addressed.

4.     The type of software used should be given.      

v Introduction

1.     Appropriate reference(s) should be cited for each paragraph in this section. For instance, the final sentences of the first paragraph and second paragraph. The second paragraph should be merged into the final paragraph of the “Introduction”.

2.     The authors should use abbreviations after addressing them in the text. For instance, CFRP.

3.     It is recommended to incorporate a paragraph, highlighting the pros and cons of the MD. The authors are suggested to provide in-detailed discussion why they employed the MD rather than other simulation models. 

4.     The authors are recommended to discuss more about the properties and applications of the thermosetting polymers.

5.     Although the authors stated the main aims of the study in the final paragraph of the “Introduction” section, it is recommended to clearly address the novelty of the work as well.  

6.     This section should be enriched by concisely providing the results of the similar works.    

v Simulation method

1.     “First, relaxation calculation is performed in the NVT ensemble at a temperature of T=600K to randomize the molecular arrangement inside the system.” Please clearly address why relaxation calculation is done in the NVT at 600 K. Is it a constant temperature or can it be varied within a specific range?   

2.     The same comment for choosing the strain rate and strain amplitude. Why did the authors not use higher strain amplitudes?

3.     The authors addressed Fig.4 after Fig. 2 (in the main text). Moreover, I couldn’t find Fig.4 in the manuscript.

v Results and discussion (rather than results and discussions)   

1.     Fig. 3 is placed on the text so that some part of the text is unreadable.

2.     There are two sentences, explaining what Fig. 4 shows, as follows: “Figure 4 illustrates the potential covalent bonds for breaking on the molecular structure.” and “Figure 4 represents stress-strain curves for the first and 10th cyclic loadings.”.

3.     The authors have done numerical simulations. However, it would be excellent if they can do some experimental tests to approve the main outcomes.

4.     Is it possible for the authors to present the data for a higher number of cyclic loadings?

5.     The authors are suggested to compare the results with those previously published and highlight the advantages of their work.

6.     It is recommended to improve the quality of the figures. 

7.     Please provide in-depth discussion on microscopic damage of thermosetting resin under cyclic loading using the published data in this field and establish a relationship between the practical and numerical results.  

8.     It is recommended to report the efficiency and accuracy of the simulation process employed.

v Conclusions (rather than conclusion)

1.     It is suggested to revise this section and systematically list the main achievements of the work.  

2.     Addressing potential application(s) can attract much more attention to the paper.   

Comments on the Quality of English Language

Moderate editing of English language required

Author Response

Replies to Referee #1

          Thank you very much for your careful reading of our manuscript and for your valuable comments.  According to your criticisms and comments, we made a revision of the manuscript.  Replies to your individual comments are listed below.

[Comment #1]

The abstract needs to be rearranged to make it easier for readers to understand the aims and objectives of this paper.

[Reply]

Thank you for the important suggestion. In response to the reviewers' comments, we have completely rewritten the abstract.

-----------------------------------------------------------

[Comment #2]

Research gaps need to be sharpened in the introduction.

[Reply]

To highlight the novelty of this study compared to previous research, the following statement has been added to the paper.

(Page 3, lines 8 from the bottom)

Previous studies have also explored the durability testing of thermosetting polymers. For example, Park et al. and Li et al. investigated the correlation between changes in the internal structure of thermosetting resins and their mechanical properties using classic MD calculations that do not involve covalent bond dissociation [48,49]. Similarly, Schichtel et al. conducts cyclic loading tests considering both bond formation and dissociation using Reax-FF [50]. They failed to accurately reproduce the fracture strain and fracture stress of real thermosetting resins, although they obtained valuable insights into the changes in the internal structure of thermosetting polymers. Our methodology, integrating cyclic loading calculations and covalent bond dissociation within classical MD simulations, represents an important step forward in quantitatively assessing microscale damage, introducing a unique novelty absent in prior studies.

-----------------------------------------------------------

[Comment #3]

The placement and size of all images need to be rearranged.

[Reply]

The resolution, size, and placement of all figures have been adjusted in the manuscript.

-----------------------------------------------------------

[Comment #4]

Figure 4 referred to in the paragraph is not found in the manuscript.

[Reply]

We apologies for the oversight and thank you very much for the suggestion. Figure 4 has been newly added to the revised manuscript.

-----------------------------------------------------------

[Comment #5]

It is necessary to state the reasons for determining simulation parameters such as strain rate, temperature and others. In the methodology there is no basis for selecting all parameters.

[Reply]

Thank you for the valuable suggestion. We have added the explanations regarding the simulation parameters' rationale to our revised manuscript.

(Page 5, lines 12)

The strain rate used in this study was determined based on the previous research by Park et al., where two different strain rates were employed for cyclic loading simulations [49]. We selected a strain rate slightly lower than the lower rate used by Park et al.  Three different strain amplitudes are applied: 0-0.02, 0-0.03, and 0-0.04, all of which are below the fracture strain range of actual thermosetting resins under uniaxial tensile tests.

(Page 5, lines 17 from the bottom)

In this study, each covalent bond is set to break when its length exceeds 1.1 times the equilibrium bond length () for the OPLS-AA force field. Preliminary validation using uniaxial tensile tests was conducted to investigate the relationship between the criteria length and the strain at which covalent bond dissociation begins. The results showed that bond dissociation initiates at a strain of approximately 5% when  is used. This value was set as the criteria length, ensuring that no covalent bond dissociation occurs up to the maximum strain amplitude of 4% used in this study.

-----------------------------------------------------------

[Comment #6]

Need a more comprehensive explanation of this sentence "Our results also show the system density decreases with the number of dissociated bonds, which is attributed to the formation of voids within the system resulting from the increased presence of free-edge atoms within the cross- linked polymers"

[Reply]

Thank you for the suggestion. We have revised the previous sentences that the reviewers pointed out to the following descriptions.

(Page 8, lines 4 from the bottom)

Our findings indicate a reduction in system density correlating with the number of dissociated bonds, signifying an expansion in system volume pre- and post-cyclic loading. This phenomenon can be attributed to the following: When covalent bonds break, the atoms within them gain mobility, leading to an increase in the entropy of center-of-mass freedom. To maximize entropy, the freedom of movement for polymer end atoms is enhanced, resulting in an expansion of system volume and, consequently, a decrease in density. This increase in volume can be represented as an increase in voids.

-----------------------------------------------------------

[Comment #7]

It is necessary to explain Figure 7, especially at the rate of change strain of 0.02. Why is the result negative? Has the cross section area increased?

[Reply]

When strain amplitude is 0.02, as depicted in Figure 6, there is little covalent bond dissociation. As a result, the cross-sectional area decreases and the system density increases before and after cyclic loading, regardless of the presence or absence of covalent bond dissociation.

We have newly added these sentences to Page 10 of our revised manuscript.

Reviewer 2 Report

Comments and Suggestions for Authors

Naoki Yamada, Mayu Morita, Maruri Takamura, Takahiro Murashima, Yutaka Oya and Jun Koyanagi “Molecular dynamics simulation for cumulative fatigue damage of a thermosetting crosslinked polymer”, this paper show the covalent bond dissociation increases with increasing both cyclic load and its amplitude. The effect of covalent bond dissociation on the deterioration of mechanical properties, entropy formation and void content is shown. However, before any decision is made on its publication, mandatory revision is required in order to clarify some points and increase its attractiveness to the general public journal polymers. See comments below.

1)      “The thermosetting resin forms a three-dimensional crosslinked structure with covalent bonds, which undergo fracture under mechanical loading conditions, leading to macroscopic material failure.” - reference is required.

2) Figure 3.  - There is no explanation in the text, and the figure itself must be positioned correctly. I assume that Figure 3 is Figure 4.

3) The drawings must be brought to a general form.

4) What a calculation error

5) The discussion could be enriched by more explicit mention of specific future research directions or unanswered questions arising from the study results. This would provide a clear path for further investigation in the field.

6) Authors should make an effort to incorporate more recent references (less than 5 years old).

7) Conclusions must be presented in the form of points

Comments on the Quality of English Language

Moderate editing of English language required

Author Response

Replies to Referee #2

          Thank you very much for your careful reading of our manuscript and for your valuable comments.  According to your criticisms and comments, we made a revision of the manuscript.  Replies to your individual comments are listed below.

[Comment #1]

“The thermosetting resin forms a three-dimensional crosslinked structure with covalent bonds, which undergo fracture under mechanical loading conditions, leading to macroscopic material failure.” - reference is required.

[Reply]

Thank you for the suggestion. We have newly added the following reference for the description.

Vassilopoulos A. P. The history of fiber-reinforced polymer composite laminate fatigue. International Journal of Fatigue, 2020, 134, 105512.

-----------------------------------------------------------

[Comment #2]

Figure 3.  - There is no explanation in the text, and the figure itself must be positioned correctly. I assume that Figure 3 is Figure 4.

[Reply]

-----------------------------------------------------------

[Comment #3]

The drawings must be brought to a general form.

[Reply]

The resolution, size, and placement of all figures have been adjusted in the manuscript.

-----------------------------------------------------------

[Comment #4]

What a calculation error.

[Reply]

Thank you for your variable suggestion. In this study, independent cyclic loading calculations are conducted along the x, y, and z directions. The error bars shown in the results for voids, entropy, and the number of covalent bond dissociations represent the standard deviation obtained from these three calculations.

Therefore, we have added above sentences to Page 5 of our revised manuscript.

-----------------------------------------------------------

[Comment #5]

The discussion could be enriched by more explicit mention of specific future research directions or unanswered questions arising from the study results. This would provide a clear path for further investigation in the field.

[Reply]
We've incorporated the following sentences into the revised manuscript to outline future research directions based on the current study.

(Page 10, lines 11)

In future perspectives, there is a need to refine the algorithms regarding covalent bond dissociation. While this study employs a uniform threshold of 1.1 times the equilibrium bond length for all types of covalent bonds, ideally, this threshold should vary depending on the bond type. Incorporating the ease of dissociation for each specific bond type requires leveraging quantum chemistry calculations, leading to more quantitative predictions of microscopic damages for various thermosetting polymers and their composite materials. Furthermore, it is important to investigate the long-term durability of thermosetting polymers through a higher number of cyclic loading tests. For this purpose, the stability of our MD simulation also needs to be improved.

-----------------------------------------------------------

[Comment #6]

Authors should make an effort to incorporate more recent references (less than 5 years old).

[Reply]

Thank you for your variable suggestion. We have added the following references to our revised manuscript.

(1) Müzel S. D., Bonhin E. P., Guimarães N. M., Guidi E. S. Application of the finite element method in the analysis of composite materials: A review. Polymers, 2020, 12, 818.

(2) Solahuddin B. A., Yahaya F. M. A state-of-the-art review on experimental investigation and finite element analysis on structural behaviour of fibre reinforced polymer reinforced concrete beam. Helion, 2023, 9 (3), el4225.

(3) Wan L., Ismail Y., Sheng Y., Ye J., Yang D. A review on micromechanical modelling of progressive failure in unidirectional fibre-reinforced composites. Composites Part C: Open Access, 2023, 10, 100348.

(4) Ratna, D. In Recent Advances and Applications of Thermoset Resins, 2nd ed.; Elsevier: Amsterdam, The Netherlands, 2022.

(5) Dhanasekar S., Stella T. J., Thenmozhi A., Bharathi N. D., Thiyagarajan K., Reddy Y. S., Srinivas G., Jayakumar M. Study of polymer matrix composites for electronics applications, Journal of Nanomaterials, 2022, 8605099.

(6) Karak N. Overview of Epoxies and Their Thermosets. Sustainable epoxy thermosets and nanocomposites Napaam, Tezpur, India. 2021. p. 1–36.

-----------------------------------------------------------

[Comment #7]

Conclusions must be presented in the form of points.

[Reply]

In accordance to the reviewer’s comment, we have completely rewritten the conclusion as follows.

(Page 10)

The accumulation of microscopic damage induced by cyclic loading is evaluated using full-atom molecular dynamics simulations that incorporate covalent bond dissociation based on interatomic distance criteria. Our results lead to the following conclusions. 

(1) Entropy and void volume increase with both the number of cyclic loads and their amplitude, resulting in the thermosetting polymer exhibiting non-elastic behavior. The accumulated entropy and void volume showed a strong positive correlation with the number of dissociated bonds.

(2) Damage at the nanoscale initiates and propagates through the following process: Initially, under cyclic loading, covalent bonds align with the applied load direction. Subsequently, tensioned covalent bonds break, causing surrounding bonds to fracture sequentially, resulting in the formation of significant voids. Consequently, stress-strain curves display non-linear and inelastic behavior.

(3) Covalent bond dissociation reduces system density and mitigates the decrease in dissipation energy per cycle, qualitatively in line with previous experimental observations.

In summary, incorporating covalent bond dissociation into molecular dynamics simulations is crucial for reproducing damage in thermosetting resins under fatigue loading. In the future, the development of a multiscale simulation model linking quantum chemistry calculations and FEM through the current MD simulations is expected. This model could accelerate the development of thermosets with long-term durability in polymer composites.

Reviewer 3 Report

Comments and Suggestions for Authors

Thank you for submitting your manuscript to this journal. The manuscript with the title "Molecular dynamics simulation for cumulative fatigue damage of a thermosetting crosslinked polymer" contains several things that need to be improved:

1. The abstract needs to be rearranged to make it easier for readers to understand the aims and objectives of this paper

2. Research gaps need to be sharpened in the introduction

3. The placement and size of all images need to be rearranged

4. Figure 4 referred to in the paragraph is not found in the manuscript

5. It is necessary to state the reasons for determining simulation parameters such as strain rate, temperature and others. In the methodology there is no basis for selecting all parameters

6. Need a more comprehensive explanation of this sentence "Our results also show the system density decreases with the number of dissociated bonds, which is attributed to the formation of voids within the system resulting from the increased presence of free-edge atoms within the cross- linked polymers"

7. It is necessary to explain Figure 7, especially at the rate of change strain of 0.02. Why is the result negative? Has the cross section area increased?

Author Response

Replies to Referee #3

          Thank you very much for your careful reading of our manuscript and for your valuable comments.  According to your criticisms and comments, we made a revision of the manuscript.  Replies to your individual comments are listed below.

[Comment #1]

General comment

The authors should use abbreviations after addressing them in the text.

There are several grammatical and typo errors in the text.

The dots at the end of sections’ titles should be removed.   

[Reply]

Thank you for the variable suggestions. We have revised the entire paper according to the reviewer's comments.

-----------------------------------------------------------

[Comment #2]

Abstract

  1. This section should be re-written. It is highly recommended to follow the standard protocol for writing “abstract”. One sentence to state the importance of the topic, followed by the explicit illustration of the aims (and the variable), employed methods, main achievements, and an outlook in the light of obtained results.
  2. The authors are recommended to add more quantitative data in this section.
  3. The type of “base resin” and “curing agent” should be addressed.
  4. The type of software used should be given.   

[Reply]

Thank you for the important suggestion. In response to the reviewers' comments, we have completely rewritten the abstract as follows.

(Page 1 in our revised manuscript)

To develop composite materials with durability, it is crucial to elucidate the correlation between nanoscale damage in thermosetting resins and the degradation of their mechanical properties. This study aims to investigate the correlation by performing cyclic loading tests on the cross-linked structure of diglycidyl ether bisphenol A (DGEBA) and 4'4-diamino diphenyl sulphone (44-DDS) using full-atomic molecular dynamics (MD) simulations. To accurately represent nanoscale damage in MD simulations, a bond dissociation algorithm based on interatomic distance criteria is applied, and three characteristics are used to quantify microscopic damage: stress-strain curve, entropy generation and formation of voids. As a result, the number of covalent bond dissociations increases with both the cyclic loading and its amplitude, resulting in higher entropy generation, void formations, causing the material to exhibit non-elastic behavior. Furthermore, our findings indicate a microscopic degradation process in the crosslinked polymer: Initially, covalent bonds align with the applied load direction. Subsequently, tensioned covalent bonds sequentially break, resulting in significant void formation. Consequently, stress-strain curves exhibit nonlinear and inelastic behavior. While our MD simulations employ straightforward criteria for covalent bond dissociation, they unveil a distinct correlation between the number of bond dissociation and micro-scale damage. Enhancing the algorithm holds promise for yielding more precise predictions of material degradation processes.

 -----------------------------------------------------------

[Comment #3]  

Appropriate reference(s) should be cited for each paragraph in this section. For instance, the final sentences of the first paragraph and second paragraph. The second paragraph should be merged into the final paragraph of the “Introduction”.

[Reply]

Thank you for the suggestion. We have rewritten many parts of the introduction section and updated the citations in response to the reviewer's comments.

 -----------------------------------------------------------

 [Comment #4]  

The authors should use abbreviations after addressing them in the text. For instance, CFRP.

[Reply]

We have made corrections to the abbreviations throughout the entire paper.

 -----------------------------------------------------------

 [Comment #5]  

It is recommended to incorporate a paragraph, highlighting the pros and cons of the MD. The authors are suggested to provide in-detailed discussion why they employed the MD rather than other simulation models.

[Reply]

We have added a paragraph to clarify the reasons for selecting MD simulation among various computational approaches as follows.

(Page 2, lines 8)

Among various computational approaches, the finite element method (FEM) and molecular dynamics (MD) simulations are prominent for addressing the damage mechanisms of polymers and polymer composites. While FEM has successfully reproduced failure and mechanical properties quantitatively [3-5], MD simulations used in this study offer certain advantages. MD simulations calculate thermodynamic properties based on atomic motion, whereas FEM relies on the constitutive law of a material expressed as a continuum body. Consequently, MD simulations allow us to examine the dependence of microscopic damage on material degradation at the atomic scale.

 -----------------------------------------------------------

 [Comment #6]

The authors are recommended to discuss more about the properties and applications of the thermosetting polymers.

[Reply]

We have added the following sentences regarding the properties and the applications of thermosetting polymers to our revised manuscript.

(Page 2, lines 17)

A thermosetting resin forms a three-dimensional crosslinked structure with covalent bonds, ensuring lighter weight, superior mechanical properties, and higher thermal and chemical resistance [6]. Therefore, thermosetting resins are widely used not only in large structural components but also in automotive parts, industrial product components, and electrical and electronic products [7,8]. However, cross-linked structures are susceptible to damage under mechanical loading conditions, resulting in brittle fracture at lower strain levels compared to thermoplastic polymers [9,10].

 -----------------------------------------------------------

 [Comment #7]

Although the authors stated the main aims of the study in the final paragraph of the “Introduction” section, it is recommended to clearly address the novelty of the work as well. This section should be enriched by concisely providing the results of the similar works.   

[Reply]

To highlight the novelty of this study compared to previous research, the following statement has been added to the paper.

(Page 3, lines 8 from the bottom)

Previous studies have also explored the durability testing of thermosetting polymers. For example, Park et al. and Li et al. investigated the correlation between changes in the internal structure of thermosetting resins and their mechanical properties using classic MD calculations that do not involve covalent bond dissociation [48,49]. Similarly, Schichtel et al. conducts cyclic loading tests considering both bond formation and dissociation using Reax-FF [50]. They failed to accurately reproduce the fracture strain and fracture stress of real thermosetting resins, although they obtained valuable insights into the changes in the internal structure of thermosetting polymers. Our methodology, integrating cyclic loading calculations and covalent bond dissociation within classical MD simulations, represents an important step forward in quantitatively assessing microscale damage, introducing a unique novelty absent in prior studies.

 -----------------------------------------------------------

  [Comment #8]

“First, relaxation calculation is performed in the NVT ensemble at a temperature of T=600K to randomize the molecular arrangement inside the system.” Please clearly address why relaxation calculation is done in the NVT at 600 K. Is it a constant temperature or can it be varied within a specific range?  

[Reply]

In response to the reviewers' comments, we have added the following sentences to the manuscript.

(Page 9, lines 9)

To ensure a uniform crosslinked structure, it is essential that the molecules are uniformly dispersed prior to the crosslinking reaction. Since DGEBA and 44-DDS are in liquid and solid states, respectively, at room temperature, the molecules are first randomized at high temperature. Thus, relaxation calculation is performed in the NVT ensemble at a constant temperature of T=600K to randomize the molecular arrangement inside the system.

 -----------------------------------------------------------

 [Comment #9]

The same comment for choosing the strain rate and strain amplitude. Why did the authors not use higher strain amplitudes?

 [Reply]

In this study, the maximum strain amplitude is set to 0.04. This is because thermosetting resins typically undergo brittle fracture at around 0.05 strain in uniaxial tensile tests. Therefore, applying cyclic loading beyond this strain would be futile.
(Page 5, lines 14)

Therefore, to explain the reasons for selecting the strain amplitude used in this study, we have added the following sentences to our revised manuscript.

Three different strain amplitudes are applied: 0-0.02, 0-0.03, and 0-0.04, all of which are below the fracture strain range of actual thermosetting resins under uniaxial tensile tests.

 -----------------------------------------------------------

 [Comment #10]

The authors addressed Fig.4 after Fig. 2 (in the main text). Moreover, I couldn’t find Fig.4 in the manuscript. Fig. 3 is placed on the text so that some part of the text is unreadable. There are two sentences, explaining what Fig. 4 shows, as follows: “Figure 4 illustrates the potential covalent bonds for breaking on the molecular structure.” and “Figure 4 represents stress-strain curves for the first and 10th cyclic loadings.”.

 [Reply]

We apologize for the oversight and thank you very much for the suggestion. We have corrected all you have pointed out. Further, Figure 4 has been newly added to the revised manuscript.

 -----------------------------------------------------------

 [Comment #11]  

Fig. 3 is placed on the text so that some part of the text is unreadable.

 [Reply]

We again apologize for the oversight. The correction has been made in our revised manuscript.

 -----------------------------------------------------------

 [Comment #12]  

The authors have done numerical simulations. However, it would be excellent if they can do some experimental tests to approve the main outcomes.

 [Reply]

Thank you for important suggestion. We are currently planning to measure microscale damage based on cyclic loading tests. In the near future, we intend to publish a study comparing the results of improved MD simulations with experimental results.

 -----------------------------------------------------------

 [Comment #13]  

Is it possible for the authors to present the data for a higher number of cyclic loadings?

 [Reply]

Currently, calculating longer cyclic tests with MD simulations is challenging. In our study, the dissociation of covalent bonds leads to computational instability, requiring a smaller time step for atomic motion compared to typical classical MD simulations. However, we believe this issue can be overcome by refining the relaxation process.

Therefore, we have included sentences on future research perspectives in the revised manuscript.

(Page 10, lines 16 from the bottom)

Furthermore, it is important to investigate the long-term durability of thermosetting polymers through a higher number of cyclic loading tests. For this purpose, the stability of our MD simulation also needs to be improved.

 -----------------------------------------------------------

 [Comment #14]  

The authors are suggested to compare the results with those previously published and highlight the advantages of their work.

 [Reply]

Thank you for the valuable comment. In order to emphasize the advantage of this MD simulation methodology, we have added the following sentences to our revised manuscript.

(Page 10, lines 10)

Therefore, our computational method, which integrates covalent bond dissociation into classical MD simulations, demonstrates superior density reproducibility compared to conventional classical MD. Additionally, it may offer advantages in computational speed and quantitative accuracy over Reax-FF-based MD simulations.

 -----------------------------------------------------------

 [Comment #15]

 It is recommended to improve the quality of the figures.

 [Reply]

The resolution, size, and placement of all figures have been adjusted in the manuscript.

 -----------------------------------------------------------

 [Comment #16]

Please provide in-depth discussion on microscopic damage of thermosetting resin under cyclic loading using the published data in this field and establish a relationship between the practical and numerical results.  

 [Reply]

Thank you for variable comment. We have revised the following discussion that compares the results of our study with previous experimental research on cyclic loading of thermosetting resins.

(Page 9, lines 7 from the bottom)

Here, we introduce the previous experiment study in cyclic loading for the same thermosetting polymer as that used in this study. Kudo et al, reported that the density decreases after 100 cyclic loadings [61], which is qualitatively consistent with our result (Figure 7). This decrease in density cannot be reproduced by standard MD simulation without considering bond dissociation [48,49]. Additionally, they evaluated entropy generation before and after 100 cyclic loadings for amplitude of 0.01. They estimated generated entropy of 18.1 kJ/m3 through changing the heat capacity. Direct comparison between their results and ours is not possible due to differences in the number of cycles and amplitude. However, for 10 cycles with an amplitude of 0.02, our calculated entropy generation is approximately 7.5 kJ/m3. Considering the dependencies of entropy on amplitude and the number of cycles, we believe there is no significant difference. They have also reported an increase in entropy generation per cycle with the number of cyclic loadings, a trend not observed in our study or any previous ones.  However, our simulations indicate that considering void dissociation suppresses the decrease in entropy generation per cycle. These results highlight the importance of bond dissociation in replicating the failure of thermoset polymers under cyclic loading conditions.

 -----------------------------------------------------------

 [Comment #17]

It is suggested to revise this section and systematically list the main achievements of the work. Addressing potential application(s) can attract much more attention to the paper. 

 [Reply]

In accordance to the reviewer’s comment, we have completely rewritten the conclusion as follows.

(Page 10)

The accumulation of microscopic damage induced by cyclic loading is evaluated using full-atom molecular dynamics simulations that incorporate covalent bond dissociation based on interatomic distance criteria. Our results lead to the following conclusions. 

(1) Entropy and void volume increase with both the number of cyclic loads and their amplitude, resulting in the thermosetting polymer exhibiting non-elastic behavior. The accumulated entropy and void volume showed a strong positive correlation with the number of dissociated bonds.

(2) Damage at the nanoscale initiates and propagates through the following process: Initially, under cyclic loading, covalent bonds align with the applied load direction. Subsequently, tensioned covalent bonds break, causing surrounding bonds to fracture sequentially, resulting in the formation of significant voids. Consequently, stress-strain curves display non-linear and inelastic behavior.

(3) Covalent bond dissociation reduces system density and mitigates the decrease in dissipation energy per cycle, qualitatively in line with previous experimental observations.

In summary, incorporating covalent bond dissociation into molecular dynamics simulations is crucial for reproducing damage in thermosetting resins under fatigue loading. In the future, the development of a multiscale simulation model linking quantum chemistry calculations and FEM through the current MD simulations is expected. This model could accelerate the development of thermosets with long-term durability in polymer composites.

Round 2

Reviewer 1 Report

Comments and Suggestions for Authors

The authors have addressed the vast majority of my comments. The revised manuscript deserves to be published in “Polymers”.

Comments on the Quality of English Language

Minor editing of English language required

Author Response

Thank you very much for your careful reading of our manuscript. I have reviewed all the English in the document again and corrected any grammar mistakes.

Reviewer 3 Report

Comments and Suggestions for Authors

1. English proofreading is highly recommended

2. Can you explain how voids are initially formed?

3. Can this method be used to predict fatigue in polymers?

4. Please explain what is the main factor in bond dissociation when the temperature is constant?

Author Response

Thank you very much for your careful reading of our manuscript and for your valuable comments.  According to your criticisms and comments, we made a revision of the manuscript.  Replies to your individual comments are listed below. 

[Comment#1] 
English proofreading is highly recommended

[Answer]
  I have reviewed all the English in the document again and corrected any grammar mistakes.

----------------------------------------------------------------

[Comment#2]
Can you explain how voids are initially formed?

[Answer]
  The correlation between the number of covalent bond dissociation over cycles (Figure 6) and the total void volume over cycles (Figure 8) indicates a strong relationship from the initial void formation to the expansion stage. Hence, it suggests that initial voids are formed due to covalent bond dissociation.

----------------------------------------------------------------

[Comment#3]
Can this method be used to predict fatigue in polymers?

[Answer]
  We believe that the method used in this study can reproduce the formation and propagation of microscale damage under cyclic loading conditions. However, as the reviewers have expressed concerns, further investigation is needed to determine whether the method can also replicate material fatigue failure. To prevent any misunderstandings among readers, we have revised the use of the term "fatigue," which was initially used to refer to cyclic loading. Consequently, we have also updated the title of the paper as follows:
  Old Title:"Molecular Dynamics Simulation for Cumulative Fatigue Damage of a Thermosetting Crosslinked Polymer "
  New TItle:"Molecular Dynamics Simulation of Cumulative Microscopic Damage in a Thermosetting Polymer Under Cyclic Loading"

----------------------------------------------------------------

[Comment#4]
Please explain what is the main factor in bond dissociation when the temperature is constant?

[Answer]
  In this MD simulation, covalent bonds are broken due to elongation induced by applied loads. 
Specifically, covalent bonds break when they reach a certain length (1.1 times their equilibrium length). The results of our MD calculations suggest the following process for covalent bond breakage:
  Initially, the orientations of the covalent bonds are isotropic. As cyclic loading is applied, the bonds align in the direction of the load and begin to bear the applied stress. This causes the covalent bonds to elongate and eventually break.
  This process of covalent bond dissociation can occur even at a constant temperature, as long as a load is applied to the covalent bonds.